# EDTA Chelation Therapy for the Treatment of Neurotoxicity

**DOI:** 10.3390/ijms20051019

**Published:** 2019-02-26

**Authors:** Alessandro Fulgenzi, Maria Elena Ferrero

**Affiliations:** Department of Biomedical Sciences for Health, University of the Study of Milan, 20133 Milan, Italy; alessandro.fulgenzi@unimi.it

**Keywords:** neurotoxicity, neurological diseases, cardiovascular diseases, EDTA

## Abstract

Neurotoxicity can be caused by numerous direct agents, of which toxic metals, organophosphorus pesticides, air pollution, radiation and electromagnetic fields, neurotoxins, chemotherapeutic and anesthetic drugs, and pathogens are the most important. Other indirect causes of neurotoxicity are cytokine and/or reactive oxygen species production and adoptive immunotherapy. The development of neurodegenerative diseases has been associated with neurotoxicity. Which arms are useful to prevent or eliminate neurotoxicity? The chelating agent calcium disodium ethylenediaminetetraacetic acid (EDTA)—previously used to treat cardiovascular diseases—is known to be useful for the treatment of neurodegenerative diseases. This review describes how EDTA functions as a therapeutic agent for these diseases. Some case studies are reported to confirm our findings.

## 1. Introduction

Damage to neurons provoked by toxic substances is defined as “neurotoxicity,” and many of its causes are reported in the present review. Certain toxic agents are able to damage neurons “directly,” specifically toxic metals, air pollution, organophosphorus pesticides, neurotoxins, chemotherapy, other drugs (including anesthesia), and pathogen infections. Indirect mechanisms of neurotoxicity that can be considered include pro-inflammatory stimulators as cytokines (interleukin-1 or IL-1, tumor necrosis alpha or TNFα, and interferon gamma or INFγ), reactive oxygen species (ROS), and adoptive immunotherapy with C19 T-cells. This review takes into consideration the direct and indirect mechanisms of neurotoxicity. However, in this context, the review describes also the relevance of ethylenediaminetetraacetic acid (EDTA) chelation therapy in removing many damages associated with neurotoxicity; it also reports some important obtained results due to this therapy.

## 2. Major Causes and Related-Mechanisms Involved in *“Direct”* Neurotoxicity

### 2.1. Toxic Metals

Twenty-one metals are now considered toxic. They are: Aluminum (Al), antimony (Sb), arsenicum (As), barium (Ba), beryllium (Be), bismuth (Bi), cadmium (Cd), cesium (Cs), gadolinium (Gd), lead (Pb), mercury (Hg), nickel (Ni), palladium (Pd), platinum (Pt), tellurium (Te), thallium (Tl), thorium (Th), tin (Sn), titanium (Ti), tungsten (W), and uranium (U). Of these, those which are mostly known to induce neurotoxicity are: Pb, Cd, Hg, Ni, and Al. Oxidative injury is linked to Pb-induced neurotoxicity, and chronic exposure to Pb can induce cognitive dysfunction. It has been shown that a lifetime exposure to Pb causes neurodegenerative damage in rats (decreased neuronal densities and brain volumes) that begins to occur during infancy and is relieved during adulthood before intensifying during old age [1]. The molecular mechanism of Pb neurotoxicity has been identified on the human neuroblastoma cell line SH-SY5Y through scherophernia-1 gene activity disruption [2]. In PC-12 cells, Cd exposure can cause significant changes in some molecules (lipids, amino acids, vitamins, organic acid, and acylcarnitine). These alterations can modulate the metabolism of energy (glycolysis, tricarboxylic acid cycle, and fatty acid β-oxidation), cell signaling, and cell membrane composition, as well as antioxidant and cellular detoxification systems [3]. Chronic low-dose exposure to inorganic Hg generates hippocampal dysfunction in rats. In fact, damages to the short- and long-term memory, cytotoxicity, and cell death through apoptosis of both astrocytes and neurons in the hippocampus have been seen [4]. Some experimental models, such as the zebrafish model, have been used to elucidate heavy-metal (Pb, Cd, and Hg) neurological toxicity and to study the functioning of the nervous system [5]. Another toxic metal, Ni, can be accumulated in the brain with or without passing through the olfactory pathway. It provokes mitochondrial damage, e.g., changes of mitochondrial membrane potential (which is a key parameter for evaluating mitochondrial function) and ATP and mtDNA concentration decreases [6]. Al has been shown to induce cell-death in neurons, neuroglia cells, and in co-cultured neural cells from newborn rats [7]. It has been identified in the brain tissue of patients affected by multiple sclerosis (MS), both at cellular and extracellular levels [8], as well as in those affected by autism [9].

We have already described the relationship between the toxic-metal burden and the onset of neurodegenerative diseases (ND), such as multiple sclerosis (MS), amyotrophic lateral sclerosis (ALS), Parkinson’s disease (PD), and Alzheimer’s disease (AD) [10].

The role of toxic metals favoring the genetic susceptibility of ND-affected patients through epigenetic modification has also already been shown. The role of Al in the initiation and progression of AD is linked to epigenetic status alterations by DNA methylation, histone modifications, and noncoding RNA [11]. The interference of Cd in DNA repair mechanisms has also been suggested [12]. Damage to DNA has also been suggested for inorganic Hg—which has been noted in human astrocytes, oligodendrocytes, and corticomotoneurons—which suggests its role in the pathogenesis of MS, ALS, and AD [13].

Particular relevance has to be reserved for Gd neurotoxicity, due to its increased clinical use.

### 2.2. Gadolinium Neurotoxicity

Magnetic resonance imaging (MRI) with gadolinium (Gd)-based contrast agents (GBCA_s_) is widely used in clinical diagnosis. Gadolinium is a heavy metal that belongs to the lanthanide family and which has to be administered to humans in a chelated form to avoid the presence of free Gd, thus reducing its toxicity. Gadolinium neurotoxicity has been demonstrated in vitro; indeed, rat cortical neurons exposed to Gd develop endoplasmic reticulum stress via oxidative injury [14]. Moreover, Gd triggers an unfolded protein response in primary cultured rat cortical astrocytes through the increased influx of extracellular Ca^2+^ [15]. Brain Gd deposition following GBCA administration was reported by Kanda et al. [16,17]. Murata et al. described Gd deposition not only in the brain but also in bone and skin [18]. Progress has been made in understanding the routes of Gd deposition in brain structures [19]. A spectrum of adverse effects and toxicity, as well as related deposition diseases, were initially described as nephrogenic systemic fibrosis [20]. In the light of these findings, the European Medicine Agency (EMA) guidelines published in 2017 recommended the withdrawal of multiple GBCAs from clinical use [21,22]; restrictions on the use of linear GBCAs in Japan have recently been adopted [23]. Though the use of macrocyclic GBCAs has induced a signal intensity increase in the dentate nucleus and the globus pallidus in children [24], results have not been confirmed in MS, except in a small number of patients [25]. Finally, in patients affected by hereditary tumor syndromes, Gd accumulation in the brain has been seen after periodical contrast-enhanced MRI screening [26].

The reported results show the link between toxic metal burden and neurotoxicity and deserve further consideration in view of adopted measures useful to prevent ND development.

### 2.3. Air Pollution

Though toxic metals can be included among the components of air pollution, other components include particulate matter (PM), ultrafine PM, gases, and organic compounds. PM, especially diesel exhaust particles (DE), is generated by traffic-related air pollution. Mice that have been acutely exposed to DE show microglia activation, increased lipid peroxidation, and neuroinflammation in various parts of the brain, e.g., the hippocampus and the olfactory bulb [27]. It has been suggested that there might be a link between exposure to traffic-related air pollution and autism spectrum disorders [28]. Recent studies show that DE can induce neuroinflammation, oxidative stress, and neurodegenerative-related tau protein overexpression and regulation by autophagy in human neuroblastoma cells [29]. Furthermore, studies using animal and cell-culture models have shown that amyloid-beta processing, anti-oxidant defense, and inflammation are altered following exposure to the constituents of polluted air that might be involved in the etiology of AD [30]. Exposure to ambient air particles decreases the expression of hippocampal glucocorticoid receptors and increases the secretion of glucocorticoids, thus activating mood-related behavioral disorders [31].

There is yet another problem that might need to be faced in the not-so-distant future. Micro- and nanosized particles are able to reach the nervous system during inhalation, avoiding the blood-brain barrier and influencing synaptic neurotransmission. The neurosafety of environmental particles, in addition to that of engineered and planetary particles, is difficult to predict. It depends on their composition; size; shape; surface properties; stability in organisms and the environment; and capability to absorb neurotoxic substances, aggregate, form dust, and interrelate with different biomolecules [32].

### 2.4. Radiation and Electromagnetic Fields

The neurotoxic effects of radiation, as well as those of electromagnetic fields, also need to be considered. Primary human astrocytes irradiated in vitro are known to develop senescence and mediate neuroinflammation and neurotoxicity [33]. Healthcare workers occupationally exposed to ionizing radiation have an increased risk of developing a variety of cancers, particularly brain cancer, that might be related to the elevated production of ROS, oxidative DNA damage, and a significant increase in the levels of interleukin-6, IL-1α, and macrophage inflammatory protein-1α [34].

Smart described the etiology of central nervous system (CNS) dysfunction in patients after irradiation as being multifactorial and influenced by age; comorbidity; psychological- and genetic-predisposition typical of the underlying malignancy; and by additional injuries caused by surgery and/or drug treatment [35]. Radiation is known to negatively affect processing speed, attention, learning and memory, retrieval, executive function, and fine motor coordination [35]. Radiation treatment can provoke side-effects including fatigue, cognitive alteration in short-term memory and concentration, pituitary dysfunction, and, on rare occasions, dementia. Radiotherapy (RT) can damage the white matter of the brain; the corpus callosum, cingulum bundle and fornix show the most prominent dose-dependent changes after RT [36]. The role of whole-brain RT to cure primary CNS lymphoma is now under discussion; it might be confined to patients unable to tolerate chemotherapy or to those for whom it has failed and have therefore undergone radiotherapy, owing to the high risk of delayed neurotoxicity after combined treatment [37]. Exposure to radiofrequency radiation causes oxidative damage to mitochondrial DNA in primary cultured neurons, explicating its neurotoxicity in the brain [38]. Moreover, exposure to extremely low-frequency electromagnetic fields induces neurotoxicity in primary cultured hippocampal neurons by activating the mitochondrial apoptotic pathway [39]. Notably, electromagnetic fields are able to enhance the cytotoxic and genotoxic effects of Gd in human lymphocytes [40].

### 2.5. Organophosphorus Pesticides

Organophosphorus pesticides (OP) cause four main neurotoxic disorders in humans: Cholinergic syndrome, intermediate syndrome, organophosphate-induced delayed polyneuropathy, and chronic organophosphate-induced neuropsychiatric disorders. In addition, links between OP and AD, OP and attention-deficit hyperactivity disorder, and OP and PD have been suggested [41]. Paraoxon, an extremely neurotoxic OP compound, increases concentration-dependent dopamine release in rats following acute intrastriatal administration [42]. Another OP, monocrotophos, influences the development of zebrafish embryos by inducing several developmental abnormalities, such as pericardial edema, altered heart development, and spinal and vertebral anomalies, in a concentration-dependent manner [43].

### 2.6. Neurotoxins

Ciguatoxin and brevetoxin, phycotoxins mainly produced by dinoflagellates gambierdiscus toxicus and Karenia brevis, provoke gastrointestinal syndrome and sensory disorders, respectively, in humans, e.g., paresthesia, pain, pruritus, and cold dysesthesia. These effects are due to voltage-gated sodium channel activation in the sensory and motor peripheral nervous systems; they also affect the immune system [44]. Botulinum neurotoxins exert their toxic effects by means of: (1) their neurospecificity; (2) the enzymatic nature of the N-terminal domain, whose activity can inactivate substrate molecules one by one; and (3) the essential role of their proteolytic substrate, i.e., soluble N-ethylmaleimide-sensitive factor attachment protein receptor (SNARE) proteins that mediate the fundamental physiologic function of neurotransmitter release at peripheral nerve terminals [45]. Unlike tetanus neurotoxins that are uniquely transported retrogradely within motor neurons, botulinum neurotoxins are also widely used as therapeutic toxins [46].

### 2.7. Chemotherapy and Other Drug Treatments

Chemotherapy-related toxicities have been noted in parallel with the development of anticancer therapies. For example, damage to the peripheral nervous system has been seen following the administration of compounds such as platinum-drugs, various antitubulins, immunomodulatory drugs, and proteasome inhibitors [47]. Even if susceptibility to oxaliplatin-induced peripheral neurotoxicity has been noted in various mouse strains, the difficulty in the identification of reliable predictors of high-risk subjects for its development and the largely incomplete knowledge of the basic mechanisms of drug-induced toxic effects on the peripheral nervous system components are critical [48].

The neurotoxic potential of lysergic acid diethylamide (LSD), initially used for psychiatric treatment, has been noted in patients in Denmark [49].

We should also remember the potential neurotoxicity of anti-tumor radiation therapy [50].

Anesthetic drugs also seem to be implicated in neurotoxicity. Perioperative nerve injury has been attributed to local anesthetic-induced neurotoxicity: Identified possible cellular mechanisms include the intrinsic caspase pathway, PI3K (phosphoinositide 3 kinase) pathway, and MAPK (mitogen-activated protein kinase) pathway [51]. Studies on the effects of volatile anesthetics in neonatal mouse models suggest one of three mechanisms for neurotoxicity: Reactive oxygen species (ROS), mediated stress and signaling, growth/nutrient signaling, and direct neuronal modulation [52].

### 2.8. Pathogens

Some neurological infections are caused by bacteria (*Meningococcal* disease) and viruses (West Nile infection, HIV-associated dementia) [53]. More recently, prion-mediated neurodegenerative disorders have been described; prions are unique pathogens that transmit biological information from one organism to another in the absence of nucleic acids [54,55]. The association between viral infection and ND has been reported, especially for the Herpes simplex virus, the Epstein Barr virus, and the Cytomegalovirus [56,57,58]. Moreover, in patients affected by MS, the role of pathogens associated with development or exacerbation of the disease include bacteria, such as *Chlamydia pneumoniae* and *Staphylococcus aureous*-produced enterotoxins that function as superantigens and virus of Herpesviridae (Epstein Barr virus and human herpes virus 6) [59]. Another study showed that MS patients in North-Eastern Poland were more likely seropositive for Epstein Barr and human Herpes virus 6 than healthy individuals [60]. Finally, the interaction between viral and environmental risk factors in the pathogenesis of MS has been reported [61].

## 3. Indirect Mechanisms of Neurotoxicity

We report both the known and the emerging related data.

### 3.1. Cytokine/ROS Production

The production of pro-inflammatory cytokines by stimulated glial cells or endothelial cells—such as IL-1, TNFα, and INFγ—can provoke neuron damage [62]. The authors found that blocking microglia activation with ethyl pyruvate or allopurinol significantly decreased axonal damage and, to a lesser extent, demyelination. Blocking TNFα significantly decreased demyelination but did not prevented axonal damage. Oxidative stress, e.g., ROS production, is a central event contributing to the degeneration of dopaminergic neurons in the pathogenesis of PD [63]. The role of nicotinamide adenine dinucleotide phosphate (NADPH) oxidases in ROS generation and the opportunities to target them in ND have been highlighted [64].

### 3.2. Adoptive Immunotherapy with CD19 T Cells

Lymphodepletion chemotherapy followed by the infusion of CD19-targeted chimeric antigen receptor-modified T (CAR-T) cells can be associated with adverse neurological effects in patients affected by refractory B-cell malignancies. Severe neurotoxicity has been related to endothelial activation (disseminated intravascular coagulation, capillary leak, and increased blood-brain barrier permeability), favoring the release/production of cytokines, including INFγ. The biomarkers of endothelial activation are higher before neurotoxicity development [65,66].

Some of the direct and indirect causes of neurotoxicity can contribute to damage humans concomitantly: As an example, ROS production is also provoked by toxic metal burden.

## 4. Which Methods Can Be Used to Detect Neurotoxicity in Humans?

Although damage to neurons or to glial cells in “in vitro” cell cultures can be assessed by testing various potentially toxic agents, neuro-intoxication in humans is difficult to detect and is based on epidemiological data. However, we can adopt the “chelation test,” which is used to reveal toxic-metal burden in humans by means of a chelating agent able to bind and remove toxic metals. Many toxic compounds, such as PM, OP, and chemotherapeutic agents, often contain toxic metals, making this method useful when assessing the damage caused by compounds other than metals. The sources and symptoms of human toxic-metal burden, the doses and routes of administration of the chelating agent in numerous human diseases, and the rationale for the use of the chelating agent calcium disodium ethylenediaminetetraacetic acid (EDTA) have already been reported extensively [67].

## 5. Toxic-Metal Detection and EDTA Chelation Therapy

The “chelation test with EDTA” was previously reported [67] and is briefly reported below. EDTA (2 g), diluted in about 500 mL physiological saline (NaCl 0.9%) is slowly (over 2 h) administered intravenously to subjects who are invited to collect urine samples before and after initial intravenous EDTA treatment. Urine collection following chelation treatment lasts 12 h. Urine samples are accurately enveloped in sterile vials and sent to the Laboratory of Toxicology (Doctor’s Data Inc., St. Charles, IL, USA) for analysis. Samples are acid-digested with certified metal-free acids (digestion takes place in a closed-vessel microwave digestion system), diluted with ultrapure water, and examined via inductively coupled plasma mass spectrometry (ICP-MS), a reliable new method for reducing interference using collision/reaction cell methods coupled with ion-molecule chemistry. Urine standards, both certified and in-house, are used for quality control and data validation. To avoid the potentially great margin of error due to fluid intake and sample volume, results are reported in micrograms (µg) per g of creatinine. When the first “chelation test” shows toxic-metal burden, patients initiate weekly chelation therapy. This consists in the intravenous slow infusion of 2 g EDTA diluted in 450 mL physiological saline once a week. Therapy duration is established following the evaluation of toxic-metal burden after almost ten subsequent therapies.

The most common toxic-metal burdens have already been assessed in a total of 1147 patients (671 affected by ND, 138 by non-ND such as fibromyalgia or other diseases, and 338 healthy controls) [10]. The toxic-metal burden of these patients was periodically monitored for more than one year. The results demonstrated that poisoned patients affected by disease were more numerous than healthy patients. Moreover, except for As, toxic metal presence was significantly more elevated in ND patients. The results underline the important relationship between the chronic body burden of some toxic metals and ND. Finally, repeated EDTA chelation therapy was able to remove all toxic metals with no adverse effects.

## 6. Beneficial Effects of Chelation Therapy with EDTA in Patients Affected by Toxic-Metal Burden

Neurological symptoms caused by drugs containing Hg were eliminated in one patient following EDTA therapy [68]. The symptoms and oxidative status in a patient affected by rheumatoid arthritis and by heavy-metal intoxication improved following EDTA therapy [69]. A patient affected by MS who had not responded to standard treatment underwent EDTA therapy because his chelation test revealed a high Al, Pb, and Hg burden. Over time, the patient showed a reduction of toxic-metal levels, and symptoms improved, thus indicating possible MS remission [70]. The involvement of Al in the pathogenesis of ND has been demonstrated; Al intoxication can be significantly reduced by short-term EDTA treatment [71] as well as by long-term treatment [72]. The antioxidant compound Cellfood, administered to patients affected by ND undergoing EDTA chelation therapy, can improve oxidative status and homocysteine metabolism [73]. EDTA chelation therapy, without added vitamin C, has been shown to decrease oxidative DNA damage and lipid peroxidation [74]. Another important link between known diseases and metal exposure has been recently identified. Indeed, chronic toxic-metal exposure is associated with cardiovascular diseases (CVD), thus indicating the role of chelation therapy in the treatment of these diseases [75]. Chelation therapy to treat CVD has been used for many years, but epidemiologic evidence linking chronic toxic metal exposure to CVD morbidity and mortality has more recently demonstrated. In particular, chelation therapy with EDTA can help prevent diabetes-associated cardiovascular events [76].

## 7. Major Mechanisms Underlying the Efficacy of EDTA Chelation Therapy against Neurotoxicity

### 7.1. Chelation of Toxic Metals

As already described above, toxic metals are known to be one of the more frequent causes of neurotoxicity. They are present in the air, food, and drugs, and, following inhalation, ingestion, or dermal absorption, can reach and damage important human organs, including the CNS [67]. EDTA is able to bind toxic metals and to create stable complexes in vitro with Pb^++^, Cd^++^, and Hg^++^ [68]. The complexes are eliminated in vivo via urine, as shown by the evaluation of toxic-metal urine levels following EDTA challenge testing.

### 7.2. Antioxidant Activity

Toxic metals are also involved in oxidative damage in cells, favoring enhanced ROS production [77]. EDTA is itself an antioxidant agent, provoking ROS reduction and the increase of total antioxidant capacity in the blood samples of Cellfood-treated ND patients [73]. It has already been shown how, after ten sessions of chelation therapy, plasma peroxide levels and DNA damage decrease significantly [74].

### 7.3. Endothelium Protection

The effects of EDTA administration were studied in a rat model of acute renal failure induced by ischemia followed by post-ischemic reperfusion. Treatment with EDTA was performed thirty minutes before ischemia induction. Functional and histological parameters were preserved by EDTA treatment. Indeed, serum creatinine levels did not increase significantly, nitric oxide (NO) levels and endothelial NO synthase renal expression improved, adhesion molecule Mac1 expression reduced, and TNFα-induced vascular leakage was prevented [78]. The in vitro effects of EDTA in modulating human umbilical vein endothelial cell (HUVEC) activation induced by TNFα were also examined. TNFα-generated F-actin stress fibers reduced, as highlighted by normal tubulin distribution, in keeping with a well spread, quiescent endothelial phenotype [79]. The results obtained regarding the avoidance of EDTA endothelial activation are in agreement with the beneficial effects of EDTA in vivo in patients affected by CVD [76,80,81].

### 7.4. Potential Antimicrobial Agent

EDTA has been shown to be potentially effective antimicrobial treatment against *Streptococcal* isolates implicated in causing bovine mastitis [82]. Tetrasodium EDTA has been proposed as an antimicrobial and antibiofilm agent for use in wound care. Indeed, a low concentration of t-EDTA (4%) solution was able to kill *Staphylococcus aureus*, methicillin-resistant *Staphylococcus aureus, Staphylococcus epidermidis, Pseudomonas aeruginosa*, and *Enterococcus faecalis* within in vitro biofilms after a 24 h contact time [83].

The proposed mechanisms of action of EDTA are shown in Figure 1 and explain the efficacy of EDTA chelation therapy in ND and CVD.

Now we report some particular case studies useful to confirm the usefulness of EDTA to chelate toxic metals. These are new cases following the same procedures than on the other reported case studies [67,68,69,70,71,72,73].

## 8. A Case-Study

A 65-year-old women from a rural Peruvian region near Lima underwent medical examination in Peru and in Italy for various clinical symptoms of an unknown origin. She arrived at our outpatient medical center in February 2018. Anamnestic data and symptoms were as follows: Loss of appetite, followed by anorexia and important body weight loss (approximately 22 kg), nausea, vomiting, constipation, abdominal pain, headache, memory loss, hypertension, fatigue, and paresthesia. The patient was unstable and revealed a complete lack of stamina. For many years she had been exposed to the organophosphorus pesticide MONOFOS, which had possibly caused the inexplicable deaths of some young men working in the same area. The patient immediately underwent chelation testing that showed intoxication due to Cd, Pb, and Hg (Figure 2), all of which are present in MONOFOS. The patient underwent chelation therapy. Symptoms improved after three applications. Nevertheless, after ten applications of chelation therapy, toxic-metal levels remained high, indicating the presence of chronic intoxication; chelation therapy was therefore continued. Following twenty chelation therapy applications, Hg levels in the urine sample following chelation testing were <4 µg/g creatinine, and the patient is now well. Symptoms have subsided, and she has recovered her body weight.

## 9. Two Cases Implying Gd Neurotoxicity

### 9.1. Case 1

A 34-year-old man was diagnosed with multiple sclerosis (MS) 13 years ago. He was initially treated with intravenous steroid therapy and thereafter underwent the following drug therapy: Interferon beta 1a (two years), glatimer acetate (one year), natalizumab (three years), fingolimod (two years), and ocrelizumab (one year). The patient also received stem-cell infusions on two separate occasions. During the same period the patient underwent 10 MRI scans, where Gd was used as a contrast medium: MRI showed numerous hyperintense surfaces (data not shown). Recently, the patient spontaneously interrupted therapy and decided to undergo the EDTA chelation test. Due to the patient’s inability to walk, he presented in a bath chair; he also had difficulty speaking. Results regarding toxic metal levels in the urine sample can be seen in Figure 3.

Notably, Gd values were found to be at levels considered unacceptable for humans. Lower amounts of the toxic metals Al, Cd, and Pb were also found. How did this patient accumulate so much Gd? Was he unable to eliminate it? Some subjects reveal the inability to detoxify themselves owing to low levels of glutathione or enzymes that help remove ROS. Was the Gd that accumulated in the patient’s brain responsible for symptom exacerbation? Was the immunosuppressant therapy associated with Gd administration the cause of the rapid deterioration of the young patient’s condition? This important result might suggest the assessment of not only renal function in patients that undergo MRI as a clinical determinant of subacute Gd toxicity.

### 9.2. Case 2

A woman was diagnosed with MS when she was 39 years old. She was treated with intravenous steroid therapy, followed by interferon beta 1a therapy for one year. The patient decided to interrupt immunosuppressant therapy owing to intolerance. She underwent chelation testing that showed Gd, Cd, and Pb intoxication, as shown in Figure 4. The patient showed significant tiredness, fine motor skills disturbance in the hands, and reduced foot sensitivity.

The patient had previously undergone only two diagnostic MRI examinations with Gd, yet this toxic metal was the most present among those found. The patient decided to undergo chelation therapy, whose beneficial effects were evident as MS symptoms disappeared. Chelation therapy was initially carried out on a weekly basis, which, after 12 months, was modified to two applications per month. However, Gd levels decreased very slowly, as shown in Figure 5, which highlights toxic metal levels in urine samples after one year of EDTA chelation therapy.

Gadolinium levels fell only after two further years of chelation treatment, as shown in Figure 6.

During therapy, all MS symptoms progressively disappeared, and the patient appeared to be in a good state of general health (EDSS = 4 before the beginning of chelation therapy; EDSS = 0 three years after). She observed correct diet avoiding glucose, took glutathione daily (250 mg, Oximix 7+ Driatec, Italy) and 15 drops of the antioxidant deutrosulfazyme three times a day (Cellfood, Eurodream, La Spezia, Italy). The patient is now well and undergoes chelation therapy twice a year.

## 10. Conclusions

Many toxic agents, as well as some pathogens, are specifically involved in neurotoxicity. Human intoxication can be prevented by a correct lifestyle, avoiding contact with toxic agents such as cigarettes, and the consumption of contaminated food. A healthy diet and physical exercise are recommended. Of note, all our patients eliminated smoking. However, in patients affected by chronic metal intoxications who have developed ND and/or CVD, the use of chelation therapy with EDTA represents an excellent option to ameliorate symptoms [84]. The relation between metal intoxication and ND is highlighted by previous studies and newly reported cases in this paper. EDTA treatments can be performed once a week without side effects for many years until the complete detoxification of the patient. The usefulness of EDTA treatment to ameliorate the symptoms of patients affected by ND deserve in our opinion more attention by the scientific community.

## Figures and Tables

**Figure 1 ijms-20-01019-f001:**
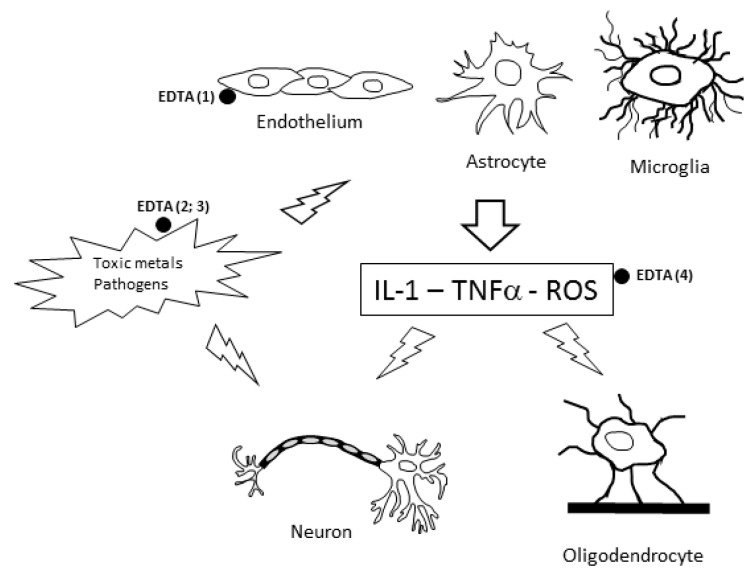
The proposed mechanisms underlying the efficacy of ethylenediaminetetraacetic acid (EDTA) chelation therapy against neurotoxicity. Toxic agents, (toxic metals, organophosphorus pesticides (OP), pathogens, air pollution, some drugs) can damage neurons and/or glial cells and endothelial cells directly (·). Activated glial cells and endothelium produce reactive oxygen species (ROS) and pro-inflammatory cytokines (IL-1, TNFα), which are able to further damage neurons. Treatment with EDTA (●) has the following effects: (1) Protection against endothelial activation; (2) removal of toxic metals; (3) possible anti-inflammatory functions (limiting pathogen infections and cytokine production); and (4) antioxidant activity (reducing ROS levels).

**Figure 2 ijms-20-01019-f002:**
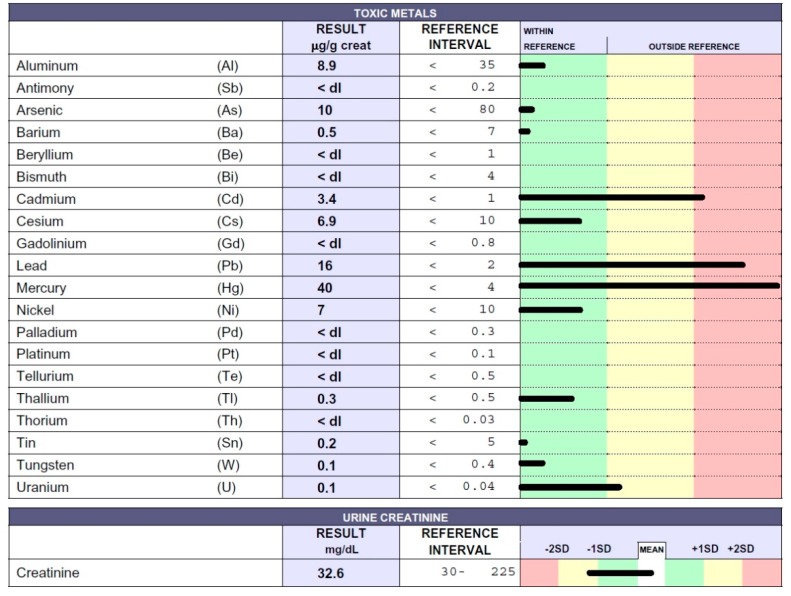
Toxic-metal levels measured by inductively coupled plasma mass spectrometry (ICP-MS) method in the patient’s urine collected during the 12 h following EDTA “challenge” (chelation test) reported in micrograms (µg) per g creatinine. The 65-year-old female patient was affected by OP intoxication.

**Figure 3 ijms-20-01019-f003:**
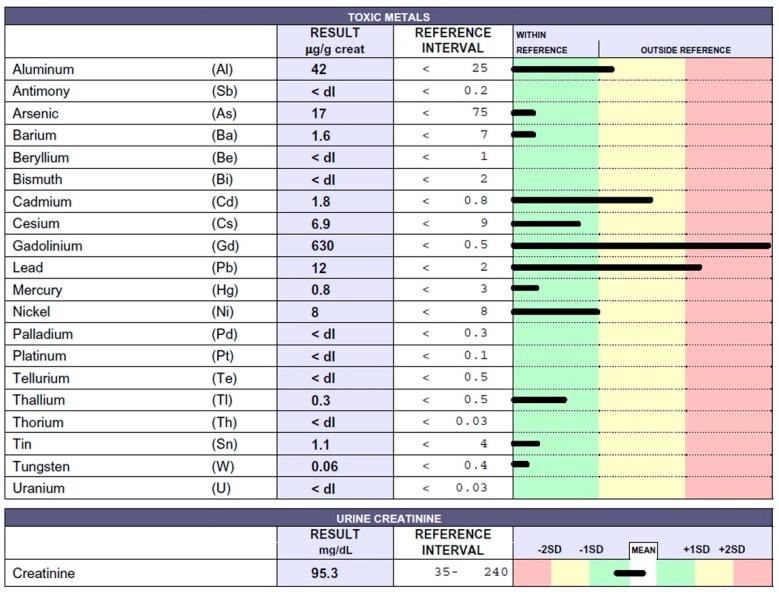
Toxic metal levels measured by ICP-MS method in the patient’s urine collected during the 12 h following EDTA chelation test reported in micrograms (µg) per g creatinine. The 34-year-old man was affected by multiple sclerosis (MS).

**Figure 4 ijms-20-01019-f004:**
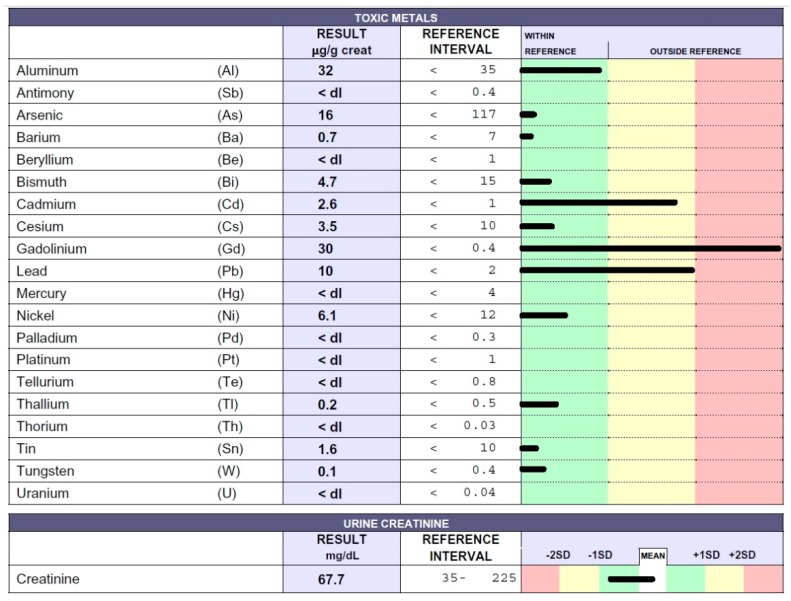
Toxic metal levels measured by ICP-MS method in the patient’s urine collected during the 12 h following EDTA chelation test reported in micrograms (µg) per g creatinine. The 39-year-old woman was affected by MS.

**Figure 5 ijms-20-01019-f005:**
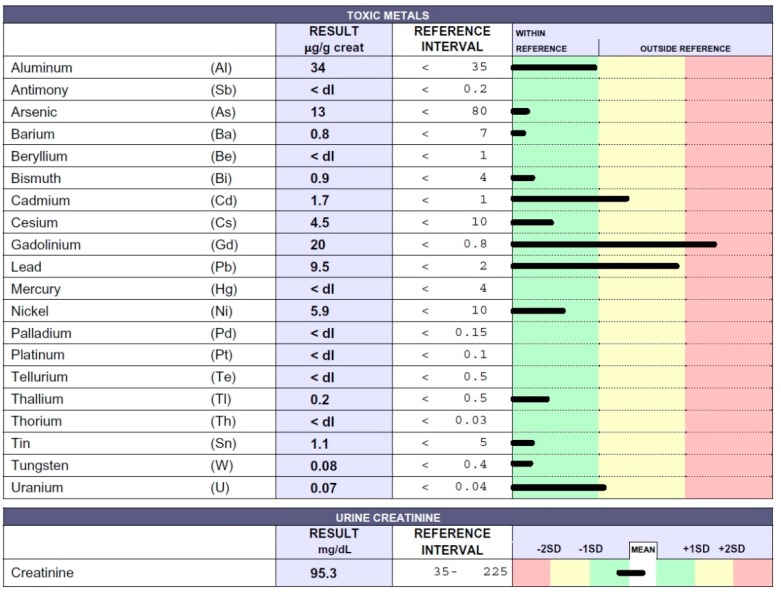
Toxic metal levels measured by ICP-MS method in the patient’s urine collected during the 12 h following EDTA chelation test reported in micrograms (µg) per g creatinine. The patient underwent chelation therapies with EDTA for a year and was affected by MS.

**Figure 6 ijms-20-01019-f006:**
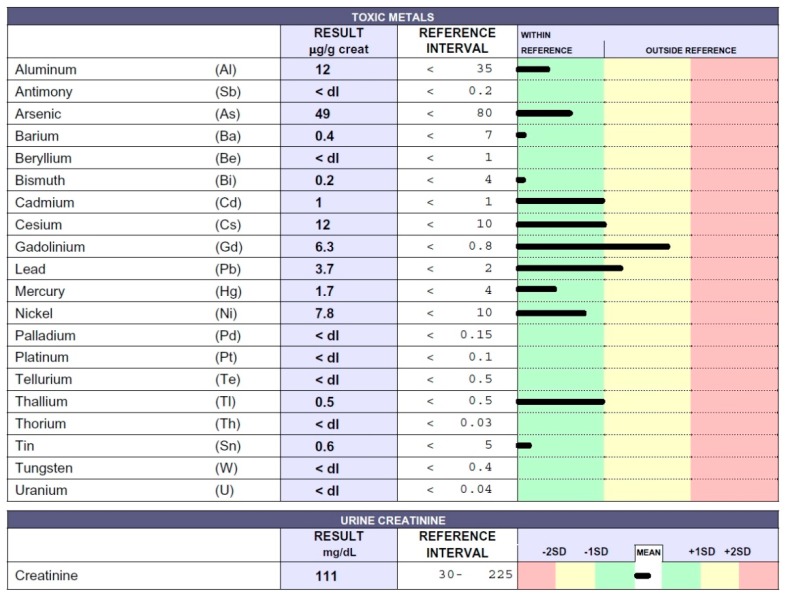
Toxic metal levels measured by ICP-MS method in the patient’s urine collected during the 12 h following EDTA chelation test reported in micrograms (µg) per g creatinine. The patient underwent chelation therapies with EDTA for three years and was affected by MS.

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
