# Peer review of "EDTA Chelation Therapy for the Treatment of Neurotoxicity"

_ijms, 2019, doi:10.3390/ijms20051019_

Round 1
Reviewer 1 Report
The manuscript "EDTA chelation therapy for the treatment of neurotoxicity" written by Fulgenzi A. and Ferrero M. E. reviews the main causes of neurotoxicity promoted by exogenous and endogenous compounds. They also present some examples of chelation therapy with EDTA and the benefits that this product has.
However, the overall manuscript does not follow a logical flow and it jumps from one topic to another without links or connections. The lack of a coherent structure of the topic make the review a bit more difficult to read. This lead to the following concerns:
Introduction.
It is very short and does not provide too much information about the context.
It doesn't define the focus of the review (is it the review about the causes of neurotoxicity or about the EDTA chelation therapy?)
It does not explain the text structure.
It has no references.
Lines 19-22: "Pro-inflammatory stimulators can be considered cytokines [...] and adoptive immunotherapy with C19 T-cells". This phrase does not make sense.
Body of the review.
Overall, is well organized in several sections but they are not well linked among them. Also, inside each section, there are only very few studies cited. The studies cited are not compared between each other and there are no critical review from the authors to the results stated in the text.
Lines 30-33: "It has been shown that lifetime exposure to Pb [...], which improves during adulthood only to intensify during old age". By "improves" the author meant that the exposure to Pb causes neurodegenerative damage during infancy but that this neurodegeneration is lower during adulthood, and then it becomes more acure during old age? It was not clear enough to me.
Line 33: "The molecular mechanism of Pb neurotoxicity has been identified on the human neuroblastoma cell line through scherophernia-1 gene activity disruption". Which neuroblastoma cell line? Please, specify it, since there are several.
Lines 38-40: Please re-arrange the two phrases, since there is no link between them and makes the reading more difficult. In general, the authors should revise this for the whole manuscript.
Lines 42-43: "Specific studies have been carried out on Pb, Cd, and Hg". Which studies? The authors only cite a review and they don't explain or review those studies.
Line 44: "It provokes mitochondrial damage at the level of membrane potential..." Please explain it better.
Lines 91-94: Is that two questions or just one? They are not well stated and they are difficult to understand.
Lines 130-131: Please double check the nomenclature of bacteria.
Lines 147-149: The authors have separated two independent phrases into two paragraphs. It seems just random information in between two sections.
Lines 157-163: The sub-section "Pathogens" should be a little bit more developed.
Line 164: First time that the authors mention the existence of indirect mechanisms of neurotoxicity in the review. They should address the structure of the review in the introduction.
Lines 166-168: The "Cytokine/ROS production" sub-sections is literally just one phrase. The authors should review the current knowledge of this well-known topic and increase the length of it.
Lines 187-202: Do the authors have any reference about the "chelation test with EDTA" that they report in the manuscript?
If yes, please appropriately cite and review it in the manuscript. A review of the suitability of a method can not be accepted with no references about that method.
If no, the authors should consider to publish the methods and the consequent results as a research article before publishing this review.
Figure 1: It does not reflect the proposed mechanism of action of EDTA. The figure is not self explanatory and the description of the figure is not clear enough.
Case studies: Are the cases shown in the manuscript published? If yes, please make sure to cite them since the authors' aim of the review is to highlight the suitability of EDTA for chelation therapy but this can not be claimed with cases reports that have not been reviewed by the scientific community.
Conclusions.
The lack of the discussion of the cited literature during the review makes necessary a section where it is discussed. Otherwise, please discuss it while citing them.
The conclusion section state a very general knowledge about lifestyle and then points out the usefulness of the EDTA therapy, but this can not be claimed without the proper references in the review body.
Author Response
Reviewer 1
Links or connections among topics have been added in the text, where it was possible. Abstract also has been improved.
Introduction. It has been improved
The focus of the review is the problem of the neurotoxicity. It is provoked by some causes. The most important causes of neurotoxicity are reported in the text. In this context the relevance of the EDTA chelation therapy in removing many damages related to neurotoxicity has been described and supported by important obtained results. The Introduction briefly reports the causes of neurotoxicity with explanation of text structure. The related references are reported in the text.
Lines 19-22 have been modified to enhance the comprehension.
All the revisions are reported in bold
Body of the review
The sections are often difficult to link among them to compare each other. The cited studies are selected among the more recently obtained due to the most advances used techniques.
The following Lines have been modified, as required by the referee:
Lines 30-33 and 33
Lines 38-40, 42-43 and 44
Line 44 and 91-94.
Lines 130-131: the nomenclature of bacteria has been checked
Lines 147-149. The second phrase has been modified
The section “pathogens” (Lines 157-163) has been improved.
The indirect mechanism of neurotoxicity is now cited in the Introduction.
Lines 166-168. The length of this sub-section has been improved.
Lines 187-202. The references 62, but also the references 63-67, describe the chelation test with EDTA.
Figure 1 It has been modified to be more clear.
Case studies. Many cases have been previously reported by us with scientific validation (see references 62-67). Now we report new cases obtained with the same validated method to confirm both the presence of metal burden and the usefulness of EDTA chelation therapy.
The properties of EDTA chelation therapy have been exhaustively discussed in the previous review (Reference 62)
Some comments have been added and the conclusion session has been improved
Reviewer 2 Report
The topic of the paper is very interesting and with the high important in clinical aspects.
Title sounds interesting.
Below are my comments:
Line 264 -there is an empty bracket
Figure 1 is not mentioned in the text of manuscript.
„In vitro” I would rather write in italics
Figure 1 : should be „oligodendrocyte”
Chapter: “Toxic-metal detection and EDTA chelation therapy”. In my opinion the description of the procedure should be implemented with propriate citations concerning the protocol.
Figures 2,3,4,5,6 in the version I got are blurred.
Figures 2,3,4,5,6 it should be necessary in my opinion to indicate within the figures captions the kind of method and the equipment used for the results obtaining.
References: 10 the title is written in uppercase
The cases described I suppose that some details i.e. concerning additional diseases, additional treatment, diet, smoking should be added. Moreover when the authors are writing about the MS the information on the EDSS should be added (before and also after therapy with EDTA).
It would be good, if the authors would take under consideration to write additional small chapter describing the perspectives for future research with the brief information – what else should be examined or improved or changed in the aspect of neurotoxicity curing using the methods described.
Are there any limitations of EDTA chelation therapy?
Author Response
Reviewer 2
The paper is a review article implemented by some case reports which permit to better understand the effectiveness of EDTA chelation therapy in removing from human body the dangerous toxic metals responsible for neurodegenerative diseases. We have modified something in the text to better coordinate the first and the other parts of the article. The modifications of the text are now reported in bold.
Case studies. Many cases have been previously reported by us with scientific validation (see references 62-67). Now we report new cases obtained with the same validated method to confirm both the presence of metal burden and the usefulness of EDTA chelation therapy.
The cardiovascular diseases are cited because recently the link between chronic toxic metal exposure and cardiovascular disease-related morbidity and mortality has been shown. In addition, EDTA chelation therapy has been used from many years to treat cardiovascular diseases.
Typos have been corrected
Figure 1 has been modified to be more informative.
Reviewer 3 Report
At first, I am confused whether this is a review article or a conglomeration of the case reports. I have a problem understanding the structure of the article. Three case reports if willing to add, should be formatted in a proper way to match the first part of the article.
Secondly, I am also confused about the title of the article is willing to explain the neurotoxicity. On the other hand, it is explaining cardiovascular diseases at the end.
Thirdly, There are several typos throughout the manuscript. This should be better checked by the authors.
Lastly and importantly, Figure 1 which suppose to transmit the important message of the article, should be more informative and convey the message in a more explicit and understandable fashion.
Author Response
Reviewer 3
Line 264: the bracket now is full.
Figure 1 is mentioned in the text (line 275).
“In vitro” is now reported in italics
Oligodendrocyte instead of oligodendrocyte is now reported in the Figure 1
Chapter: toxic metal detection and EDTA chelation therapy. The reference 62 (now reported) describes exhaustively the procedure and is also reported here.
The method used for the evaluation of toxic metals present in the urine samples (ICP-MS) has now reported in the figure legends.
Reference 10 has been revised
For the patient with MS ; the EDSS values before and after chelation therapies have been reported.
Some additional data regarding SM patient and some other informations regarding the future of the therapy have been added.
No limitations are known for EDTA chelation therapy when correctly used (see conclusions).
Round 2
Reviewer 1 Report
Introduction
It is still very poor and short for a review, yet some minor improves have been made. No references have been added to the text (ie. lines 19-20: "Damage to neurons provoked by toxic substances is defined as “neurotoxicity”, and many of its causes have already been reported in the literature." The authors claim something is on the literature but they don't cite it.)
Body of the review
Some lines have been modified appropriately. However, there is still a lack of comprehensive discussion of the few articles that are cited. The authors only mention the bibliography but they don't discuss it.
The nomenclature of bacteria has not been checked and there are still some names not in italics.
In the revised manuscript that I was provided with, the references 62-67 are related with the indirect mechanisms of neurotoxicity and not with research articles demonstrating the usefulness of the EDTA chelation method.
Case studies
In the revised manuscript that I was provided with, the references 62-67 are not published case studies in humans but in vitro studies or recent reviews. I don't doubt about the case studies or the usefulness of the EDTA chelation method, but I believe that the authors should appropriately document their findings with the right literature. The authors should either cite other case studies in humans where the same methods have been applied, and clarify in the text that these are new cases following the same procedures than on the other case studies; or they should directly publish the case articles as new research data, but not as a review article.
Conclusions
The authors claim that "The properties of EDTA chelation therapy have been exhaustively discussed in the previous review (Reference 62)". The discussion of the literature and the findings of a research article in 2013 doesn't exempt the authors to not appropriately discuss the recent literature after 6 years since that publication for a review of the same topic.
The conclusions stated in this review, regardless of their veracity, are not well supported in this review.
Author Response
Reviewer 1
Introduction.
Lines 19-20 have been modified. We consider incongrous to cite in the Introduction the same references reported in the text and related to the different causes of neurotoxicity.
All the revisions are reported in bold
Body of the review
The discussion of bibliografy (which reports only the causes of neurotoxicity) is not the priority of this review, which, at difference with previoulsy reported articles, is able to suggest clinical solutions for human intoxications.
The name of bacteria are now reported in italics.
The number of reported references (62-67) was a mistake, because related to the previous version of the review. We regret for the mistake! The correct number of references was: 67-73, as reported correctly in the new version of the review.
Case studies.
The same mistake regarding the cited references was performed in the reply to the reviewer, regarding case studies. Many cases have been previously reported by us with scientific validation (see references 67-73). Now we report new cases obtained with the same validated method to confirm both the presence of metal burden and the usefulness of EDTA chelation therapy. The phrase suggested by the reviewer has been added in the text.
Conclusions
The properties of EDTA chelation therapy have been exhaustively discussed in the previous review (reference 67 instead of 62).
Reviewer 3 Report
The author has now clearly answered the points. Thus, the manuscript has improved.
Author Response
We have revised English Language.